# Transcriptomic and Proteomic Analysis of CRISPR/Cas9-Mediated *ARC*-Knockout HEK293 Cells

**DOI:** 10.3390/ijms23094498

**Published:** 2022-04-19

**Authors:** Yu-Yuan Wang, Shih-Hsin Hsu, Hsin-Yao Tsai, Fu-Yu Cheng, Min-Chih Cheng

**Affiliations:** Department of Psychiatry, Yuli Branch, Taipei Veterans General Hospital, Hualien 98142, Taiwan; m123ay22@gmail.com (Y.-Y.W.); filvhsu@gmail.com (S.-H.H.); ashleytsai0808@gmail.com (H.-Y.T.); op19910911@gmail.com (F.-Y.C.)

**Keywords:** ARC, CRISPR/Cas9, PSD95, HSPA1A, transcriptomics, proteomics

## Abstract

Arc/Arg3.1 (activity-regulated cytoskeletal-associated protein (ARC)) is a critical regulator of long-term synaptic plasticity and is involved in the pathophysiology of schizophrenia. The functions and mechanisms of human ARC action are poorly understood and worthy of further investigation. To investigate the function of the *ARC* gene in vitro, we generated an *ARC*-knockout (KO) HEK293 cell line via CRISPR/Cas9-mediated gene editing and conducted RNA sequencing and label-free LC-MS/MS analysis to identify the differentially expressed genes and proteins in isogenic *ARC*-KO HEK293 cells. Furthermore, we used bioluminescence resonance energy transfer (BRET) assays to detect interactions between the ARC protein and differentially expressed proteins. Genetic deletion of *ARC* disturbed multiple genes involved in the extracellular matrix and synaptic membrane. Seven proteins (HSPA1A, ENO1, VCP, HMGCS1, ALDH1B1, FSCN1, and HINT2) were found to be differentially expressed between *ARC*-KO cells and *ARC* wild-type cells. BRET assay results showed that ARC interacted with PSD95 and HSPA1A. Overall, we found that ARC regulates the differential expression of genes involved in the extracellular matrix, synaptic membrane, and heat shock protein family. The transcriptomic and proteomic profiles of *ARC*-KO HEK293 cells presented here provide new evidence for the mechanisms underlying the effects of ARC and molecular pathways involved in schizophrenia pathophysiology.

## 1. Introduction

Arc/Arg3.1 (activity-regulated cytoskeleton-associated protein (*ARC*)) is an immediate early gene that is enriched in neuronal dendritic spines [1,2]. ARC mRNA and protein are rapidly induced in response to learning behaviors, and ARC is a modulator of long-term synaptic plasticity and memory consolidation [3]. It interacts with the N-methyl-D-aspartate receptor (NMDAR) complex and is a critical effector molecule downstream of many signaling pathways for long-term potentiation and depression of synaptic transmission [4,5]. ARC also functions cellular as an activity-dependent regulator of endocytic internalization of α-amino-3-hydroxy-5-methyl-4-isoxazolepropionic acid receptor (AMPAR) through interactions with dynamin and endophilin in the dendritic spine [6,7]. Additionally, ARC plays a role in actin cytoskeletal dynamics in dendritic spines [8,9]. Thus, ARC is a hub protein that interacts with NMDAR-related complex proteins, functions in AMPAR trafficking and actin regulation, and with its associated molecules, may be involved in neuron development and synaptic plasticity.

Plath et al. [10] showed that *ARC*-knockout (KO) mice fail to form long-lasting memories in implicit and explicit learning tasks but retain their short-term task acquisition memories. Penrod et al. [11] found that *ARC*-deficient mice showed reduced anxiety-like behavior, deficits in social novelty preference, and antidepressive-like behavior. Additionally, genetic disruption of *ARC* is known to recapitulate schizophrenia-relevant behavioral abnormalities [12]. In humans, ARC dysregulation contributes to various neurological and cognitive disorders, including schizophrenia [13,14], autism [15], fragile X syndrome [16], and Alzheimer’s disease [17]. Two exome sequencing-based studies found mutations in genes encoding NMDAR-related and ARC-associated proteins in schizophrenia [18,19]. Hence, *ARC* may be associated with schizophrenia symptoms, interact with other known schizophrenia genes, and demonstrate schizophrenia-related biological plausibility. ARC is also known to regulate synaptic functions through interactions with various proteins, including Presenilin [20], CaMKII [21], and Triad3A [22]; nevertheless, the functions and action mechanisms of human ARC remain poorly understood and are worthy of further investigation.

Human embryonic kidney 293 (HEK293) cells are straightforward to grow in culture and suitably used as hosts for transfection. To investigate the mechanism underlying the effects of ARC, we used the CRISPR/Cas9 genome editing system to create an isogenic *ARC*-KO HEK293 cell line and conducted RNA sequencing and LC-MS/MS analysis to identify the differentially expressed genes (DEGs) and proteins in these cells. We also used bioluminescence resonance energy transfer (BRET) assays to detect interactions between the ARC protein and the two differentially expressed proteins (PSD95 and HSPA1A). Overall, our transcriptomic and proteomic analyses revealed several ARC-regulated genes and proteins involved in the extracellular matrix (ECM), synaptic membrane, and heat shock protein family.

## 2. Results

### 2.1. Generation of Isogenic ARC-KO HEK293 Cell Lines via CRISPR/Cas9 Editing

Isogenic *ARC*-KO HEK293 cell lines were generated using CRISPR/Cas9 genome editing of HEK293 cell lines. First, guide RNA (gRNA) was designed to target unique sequences present in the 5′UTR-exon 1 of *ARC* (Figure 1A) according to a CHOPCHOP design (http://chopchop.cbu.uib.no/; last accessed on 13 January 2018). Six potential off-target sites for *ARC* gRNA were predicted using the online design tool (Appendix A), and these sites were not located in the coding regions of the known genes. A pCas-guide vector containing gRNA was transfected into HEK293 cells. After single-cell isolation, two clones with (i) a 26-bp (*ARC*:c.-24-c.2delGCCTGCCGGAGCACCTGCGCACAGAT) and (ii) a 12-bp (*ARC*:c.-23-c.-12delCCTGCCGGAGCA) homozygous deletion were identified. The edited cell lines were confirmed using Sanger sequencing (Figure 1B) and immunoblotting (Figure 1C), and the clone with the 26-bp deletion was confirmed to cause downregulation of ARC (i.e., *ARC*-KO) in HEK293 cells; thus, it was used for subsequent RNA sequencing and LC-MS/MS analysis.

### 2.2. RNA Sequencing of the ARC-KO HEK293 Cell Line

Six samples from three biological replicates (three *ARC*-KO versus three *ARC*-wild-types (WT)) were used for RNA sequencing analysis. The number of reads per sample varied from 40,411,842 to 46,470,614 among the six sequenced RNA samples (Appendix A). Principal component analysis of *ARC*-KO and *ARC*-WT libraries was used to determine data clustering based on *ARC* expression. All biological replicates of *ARC*-KO and *ARC*-WT samples were distributed in two distinct groups (Figure 2A). Differential gene expression analysis revealed 411 DEGs, including 171 and 240 downregulated and upregulated genes, respectively, for which fold change was >2, and the adjusted *p*-value was <0.05. (Figure 2B and Appendix A).

According to gene ontology (GO) enrichment analysis (adjusted *p* < 0.001; Figure 2C), several DEGs were associated with the following GO terms: extracellular matrix structural constituent (GO:0005201), collagen-containing extracellular matrix (GO:0062023), extracellular matrix organization (GO:0030198), extracellular structure organization (GO:0043062), and synaptic membrane (GO:0097060) (Table 1). The DEGs were also analyzed using STRING (version 11.5) to construct a protein–protein interaction (PPI) network, in which fibronectin 1 (FN1) was the top one-degree protein (degree = 25; adjusted *p* = 9.63 × 10^−8^; Figure 2D).

To verify RNA sequencing data, 12 DEGs (*CDH8*, *CHRM3*, *CNTN1*, *GABRE*, *LHFPL4*, *LIN7A*, *SRPX2*, *GABRB3*, *GRM8*, *SHANK2*, *LRRC7*, and *SYT11*) associated with the synaptic membrane were selected, and their mRNA expression levels were verified in biologically replicated HEK293 cells using a real-time quantitative PCR (RT-qPCR) assay. Figure 3 shows the fold changes of these genes between *ARC*-KO and *ARC*-WT cells.

### 2.3. Proteomes of ARC-KO HEK293 Cells Identified Using Label-Free Mass Spectrometry-Based Shotgun Proteomics

We used a label-free LC-MS/MS shotgun proteomics approach to identify proteins differentially expressed in *ARC*-KO HEK293 cells. Six samples from three biological replicates were analyzed (three *ARC*-KO versus three *ARC*-WT). Normalized peptide-spectrum matches (PSMs) were obtained using the following formula: (PSM in sample A/total PSM in sample A) × average PSM in all six samples. Appendix A shows the number of identifications in each sample. Appendix A shows the protein accession numbers, descriptions, and normalized PSMs of the identified proteins in the six samples. Student’s *t*-tests were used to assess differences in the normalized PSMs between the two groups at *p* < 0.01, and 72 differentially expressed proteins were identified in *ARC*-KO cells (Appendix A). According to GO enrichment analysis (Benjamini *p* < 0.001) using DAVID (the database for annotation, visualization, and integrated discovery), several differentially expressed proteins were associated with the following GO terms: cytosol (GO:0005829), extracellular exosome (GO:0070062), focal adhesion (GO:0005925), cytoplasm (GO:0005737), nucleoplasm (GO:0005654), and RNA binding (GO:0003723) (Table 2).

Ten proteins differentially expressed between the groups were selected for verification in independent biologically replicated HEK293 cells using immunoblot analysis (Figure 4A). Fold differences in the expression of these proteins in *ARC*-KO and *ARC*-WT cells were determined (Figure 4B), and seven differentially expressed proteins were confirmed: HSPA1A, ENO1, VCP, HMGCS1, ALDH1B1, FSCN1, and HINT2.

### 2.4. Immunocytochemistry and Protein–Protein Interactions

We investigated the interactions between ARC and two other proteins, PSD95 and HSPA1A, by assessing protein–protein interactions in cultured cells using immunocytochemistry and BRET assays. Immunocytochemistry revealed that ARC colocalized with PSD95 (Figure 5A up-row and Appendix A) and HSPA1A (Figure 5B up-row and Appendix A), while ARC without the spectrin domain did not colocalize with PSD95 (Figure 5A down-row) and HSPA1A (Figure 5B down-row). Moreover, BRET assays results showed that ARC^WT^ protein interacted with PSD95 and HSPA1A, whereas the interactions between ARC without the spectrin domain (ARC^w/o spectrin^) and both PSD95 and HSPA1A were abolished (Figure 5C).

## 3. Discussion

### 3.1. Genetic Deletion of the Human ARC Gene Disturbs Several Signaling Pathways

Previous sequencing studies have revealed the association between rare coding variants of ARC and NMDAR postsynaptic protein complexes and schizophrenia [18,23,24]. Because ARC is a critical effector molecule and functions downstream of many signaling pathways, ARC dysfunction could be a nexus point for synaptic dysfunction in psychiatric diseases, especially schizophrenia. To further investigate ARC in vitro, we produced an isogenic human *ARC*-KO HEK293 cell line using CRISPR/Cas9 genomic editing. Our transcriptomic and proteomic analysis of this cell line showed that genetic deletion of *ARC* disturbs several signaling pathways, including those related to the ECM, synaptic membrane, cytosol, extracellular exosome, focal adhesion, cytoplasm, nucleoplasm, and RNA binding.

### 3.2. ECM Abnormalities in ARC-KO HEK293 Cells

According to our RNA sequencing and GO analyses, the DEGs in *ARC*-KO cells were associated with four ECM GO terms (extracellular matrix structural constituent, collagen-containing extracellular matrix, extracellular matrix organization, and extracellular structure organization). ECM molecules, derived from neurons and glial cells, are secreted and accumulate in the extracellular space [25]. Brain ECM-associated molecules are involved in synaptogenesis and GABAergic, glutamatergic, and dopaminergic neurotransmission [25]. In schizophrenia, the pathophysiological dysregulation of brain ECM-related proteins, which are involved in neuronal migration, proliferation and differentiation, regulation of neurodevelopment, neuroplasticity, axon guidance, and neurite outgrowth, has been proposed, suggesting that the brain ECM and its components are potential pharmacological targets for new schizophrenia therapies [26,27,28,29]. Using STRING PPI analysis, we found that an ECM molecule, FN1, was the top one-degree hub protein identified from the related DEGs in *ARC*-KO HEK293 cells. Fibronectin mediates neurite outgrowth and axonal regeneration [30]. In fibroblasts from schizophrenic patients, fibronectin shows decreased adhesiveness and altered cellular distribution [31,32]. Thus, disrupted interactions between ARC and ECM molecules, especially FN1, may contribute to the pathophysiology of schizophrenia. Thus, it would be worthwhile investigating whether ARC also regulates ECM molecules in other psychiatric diseases.

### 3.3. Synaptic Membrane Genes in ARC-KO HEK293 Cells

Mutations in the postsynaptic ARC/NMDAR complex that disrupt interactions of proteins with NMDARs are reported to be involved in schizophrenia pathogenesis [18,19]. It is not yet clear which synaptic molecules interact with ARC during synaptic plasticity at excitatory synapses. In the present study, we identified 24 DEGs associated with the synaptic membrane in *ARC*-KO HEK293 cells. Notably, some of these DEGs have been implicated in schizophrenia pathophysiology. For example, Peykov et al. [33] found an increased burden of rare *SHANK2* missense variants in patients with schizophrenia compared with controls. Additionally, ultra-rare variants with loss of function in *GRIN2C* may increase the possible risk of schizophrenia [34]. Furthermore, downregulation of *GABRB3* may contribute to the pathophysiology and clinical manifestation of schizophrenia through altered oscillation synchronization in the superior temporal gyrus [35]. Moreover, *GRM8* has been associated with schizophrenia in the Han Chinese population [36]. The present immunocytochemistry and BRET assay results demonstrated that ARC interacted with PSD95, a scaffolding protein that is a major ARC binding protein. In a previous study, genetic variants of ARC–PSD95 were associated with cognitive function [37]. Taken together, these findings suggest that ARC-regulated synaptic membrane genes could be involved in synaptic dysfunction in schizophrenia pathogenesis.

### 3.4. Interaction between ARC and HSPA1A

According to our label-free LC-MS/MS shotgun proteomics analysis, a heat shock protein 70 (HSP70) family member, HSPA1A, was the most significantly downregulated protein in *ARC*-KO HEK293 cells. Moreover, immunocytochemistry and BRET assays revealed that HSPA1A interacted with ARC. Additionally, RNA sequencing results showed that three members of HSP70, namely, HSPA1A, HSPA1B, and HSPA1L, were significantly downregulated in *ARC*-KO HEK293 cells (Appendix A). These three proteins encode 70 kDa heat shock proteins that stabilize existing proteins against aggregation and regulate the folding of newly translated proteins in cytoprotective and cellular repair mechanisms [38]. Interestingly, HSP70 proteins are critical regulators in neurodevelopmental processes and contribute to synaptic neuroprotective events [39,40]. Moreover, Park et al. [41] showed that ARC is significantly induced in cultured cells in response to heat shock. Thus, ARC may interact with HSP70 proteins, and the association of ARC with synaptic functions and neurodevelopmental processes may be related to the pathophysiology of schizophrenia.

The genes encoding HSPA1A, HSPA1B, and HSPA1L are located in the MHC class III region of 6p21.3–22.1, which is a region implicated in susceptibility to schizophrenia [42]. Additionally, higher levels of anti-HSP70 antibodies have been found in patients with schizophrenia [43,44], and HSP70 gene polymorphisms have been associated with schizophrenia [45,46,47,48,49]. Hence, the genes encoding HSP70 family proteins might be involved in the neurodevelopmental mechanism of schizophrenia and could represent candidates for schizophrenia therapy.

### 3.5. Limitations

First, although HEK293 cells are used widely as a model cell line for cell biology research, further investigation of the functions and mechanisms of ARC in primary neuronal cells or disease models is needed to consolidate the current findings. Second, immunoblotting and RT-qPCR assay confirmed the DEGs and proteins in isogenic ARC-KO HEK293 cells for only a small subset. Hence, to obtain more insight into the molecular mechanism of ARC, we need to verify other genes and proteins in the future. Third, we only used immunocytochemistry and BRET assays to detect interactions between the ARC protein and differentially expressed proteins. The current findings must be interpreted with caution. To understand ARC function critically, characterizing protein–protein interactions through methods such as co-immunoprecipitation should be performed in future studies.

## 4. Materials and Methods

### 4.1. CRISPR/Cas9-Directed Genome Editing of Isogenic HEK293 Cell Lines

A pCas-Guide vector carrying the gRNA guide sequence (CCATCTGTGCGCAGGTGCTC) was generated using the protocol of the manufacturer (Origene, Rockville, MD, USA). Briefly, two oligos were annealed using the following conditions: 94 °C for 4 min, 75 °C for 5 min, 65 °C for 15 min, and 25 °C for 20 min. After annealing, double-stranded oligos were ligated into the precut pCas-Guide vector, and the ligation mixture was transformed into DH5α-competent cells. Plasmids were isolated and sequenced using CF3 primer (5′-ACGATACAAGGCTGTTAGAGAG-3′) to confirm the presence of the gRNA guide sequence. HEK-293 cells (ATCC: CRL-1573) were transfected with CRISPR plasmids using Lipofectamine 3000 (Invitrogen). Cells were harvested 1 week after transfection, and genomic DNA was extracted for use in genomic PCR (primer sequences: 5′-AGCGACAGACAGGCGCTC-3′ and 5′-ATCTGCAGGATCACGTTGG-3′) and T7 endonuclease assays.

### 4.2. Single CRISPR/Cas9-Edited Cell Isolation

Single CRISPR/Cas9-edited cells were isolated using a QIAscout device (QIAGEN, Germantown, MD, USA) according to the manufacturer’s protocols. Briefly, CRISPR/Cas9-edited cells were seeded and cultivated in the supplied medium on a QIAscout array (QIAGEN), and a release device containing a release needle was placed on a microscope objective. The QIAscout array was placed on the microscope stage, and a microraft containing a single cell was identified. This microraft was pierced and transferred to a 384-well plate using a magnetic wand. The collected cells were processed for further cultivation and clonal expansion. Genomic DNA was then extracted from the clonally expanded cells using PDQeX Nucleic Acid Extractor (MicroGEM, Southampton, UK). This genomic DNA was used for PCR amplification and subsequent Sanger sequencing to identify correctly edited cells.

### 4.3. Total RNA Preparation, Transcriptome Sequencing, and RT-qPCR

Total RNA from each sample was purified using a Monarch Total RNA Miniprep Kit (New England BioLabs, Ipswich, MA, USA). RNA purity and quantification were checked using SimpliNano™ Biochrom Spectrophotometers (Biochrom, Holliston, MA, USA). RNA degradation and integrity were monitored using a Qsep 100 DNA/RNA Analyzer (BiOptic Inc., Taiwan). Library preparation and RNA sequencing were completed using three biological replicates at Biotools Microbiome Research Center, Taiwan. Sequencing libraries were generated using a KAPA mRNA HyperPrep Kit (KAPA Biosystems, Roche, Basel, Switzerland) following the manufacturer’s instructions. Sequencing was performed using an Illumina NovaSeq 6000 platform. DEG analysis was performed in R using DESeq2. The obtained *p*-values were adjusted using the Benjamini and Hochberg approach to control the false discovery rate (FDR). GO enrichment analysis of DEGs was conducted using clusterProfiler (v3.10.1). PPI networks were generated and analyzed using STRING v11.5 [23]. RT-qPCR assays were performed according to a standard protocol established in our laboratory [24].

### 4.4. Protein Sample Preparation and LC-MS/MS Analysis

To prepare protein samples, cells were washed twice with cold phosphate-buffered saline (PBS) and resuspended in lysis buffer containing 20 mM HEPES (pH 7.6), 7.5-mM NaCl, 2.5 mM MgCl2, 0.1 mM EDTA, 0.1% TritonX-100, 0.1 mM Na3VO4, 50 mM NaF, and protease inhibitor cocktail (one mini-tablet/10 mL; Roche Diagnostics GmbH). The homogenates were centrifuged at 13,000 rpm and 4 °C for 30 min, and the supernatants were stored at −80 °C until they were used.

Protein solutions were first diluted in 50 mM ammonium bicarbonate (ABC) and reduced with 5 mM dithiothreitol (Merck) at 60 °C for 45 min, after which they were subjected to cysteine-blocking using 10 mM iodoacetamide (Sigma-Aldrich) at 25 °C for 30 min. Samples were then diluted with 25 mM ABC and digested using sequencing-grade modified porcine trypsin (Promega, Madison, WI, USA) at 37 °C for 16 h. The digested peptides were diluted in HPLC buffer A (0.1% formic acid) and loaded onto a reverse-phase column (Zorbax 300SB-C18, 0.3 × 5.0 mm; Agilent Technologies, Santa Clara, CA, USA). The desalted peptides were then separated on a homemade column (Waters BEH 1.7 μm, 100 μm I.D. × 10 cm with a 15 μm tip) using a multistep gradient of HPLC buffer B (99.9% acetonitrile/0.1% formic acid) for 70 min with a flow rate of 0.3 μL/min. The LC apparatus was coupled with a 2D linear ion trap mass spectrometer (Orbitrap Elite ETD; Thermo Fisher Scientific) operated using Xcalibur 2.2 software (Thermo Fisher Scientific). Full scan MS was conducted in the Orbitrap system from 400 to 2000 Da and with a resolution of 120,000 at *m*/*z* 400. Internal calibration was performed using the ion signal of protonated dodecamethylcyclohexasiloxane ions with *m*/*z* 536.165365 as the lock mass. Twenty data-dependent MS/MS scan events were followed by one MS scan for the twenty most abundant precursor ions in the preview MS scan. The *m*/*z* values selected for MS/MS were dynamically excluded for 40 s with a relative mass window of 15 ppm. The electrospray voltage was set to 2.0 kV, and the capillary temperature was set to 200 °C. MS and MS/MS automatic gain control was set to 1000 ms (full scan) and 200 ms (MS/MS) or to 3 × 106 ions (full scan) and 3000 ions (MS/MS) for maximum accumulated time or ions, respectively.

The data were analyzed using Proteome Discoverer software (version 1.4; Thermo Fisher Scientific, Waltham, MA, USA). The MS/MS spectra were searched against the SwissProt database using the Mascot search engine (version 2.5; Matrix Science, London, UK). For peptide identification, 10 ppm mass tolerance was permitted for intact peptide masses, and 0.5 Da was permitted for CID fragment ions with an allowance of two missed cleavages made from the trypsin digestion: oxidized methionine and acetyl (protein N-terminal) as variable modifications and carbamidomethyl (cysteine) as a static modification. PSMs were then filtered based on high confidence levels and Mascot search engine “rank 1” for peptide identification to ensure that the overall FDR was <0.01. Additionally, proteins with a single peptide hit were removed from the dataset.

The differentially expressed proteins (normalized PSMs between the two groups at *p* < 0.01) were subjected to the DAVID database (DAVID 2021, https://david.ncifcrf.gov/content.jsp?file=release.html; last accessed on 14 April 2022) for GO enrichment analysis.

### 4.5. Immunoblotting Assay

Immunoblotting was performed according to standard protocols using the following primary antibodies: rabbit anti-ARC (66550-1-Ig; Proteintech, Rosemont, IL, USA), rabbit anti-ENO1 (A1033; ABclonal, Woburn, MA, USA), rabbit anti-ENO2 (A3118; ABclonal), rabbit anti-FLNC (A13018; ABclonal), rabbit anti-HSPA1A (A0284; ABclonal), rabbit anti-VCP (A1402; ABclonal), rabbit anti-ALDH1B1 (A3725; ABclonal), rabbit anti-HMGCS1 (A3916; ABclonal), rabbit anti-FSCN1 (A1904; ABclonal), rabbit anti-TUBB (ab6046; Abcam), rabbit anti-HINT2 (NBP1-86024; Novus Biologicals), and mouse anti-GAPDH (G8795; Sigma-Aldrich), Saint Louis, MO, USA. Horseradish peroxidase-conjugated donkey anti-rabbit IgG (NA934V, GE Healthcare Life Sciences, UK) and human antimouse IgG (5220-0341; KPL) were used as secondary antibodies. Chemiluminescence was visualized using an enhanced chemiluminescence detection system (GTX400006; GeneTex). Immunoblot intensity was assessed using NIH ImageJ software (http://rsb.info.nih.gov/nih-image/, last accessed on 14 April 2022).

### 4.6. Construction of Expression Plasmids, Transfection, and Immunocytochemistry

The pcDNA3.1/C-terminal-green fluorescent protein (GFP) vectors containing ARC^WT^ and ARC^w/o spectrin^ were cloned following procedures described previously [25]. *DLG4* and *HSPA1A* cDNAs were cloned into expression vectors (RFP-tagged) using the PrecisionShuttle Vector System (OriGene).

COS-1 cells (ATCC: CRL-1650) were cultured in Dulbecco’s modified Eagle’s medium supplemented with 1 mM sodium pyruvate, 0.1 mM nonessential amino acids, 100 U penicillin, 100 μg streptomycin (Invitrogen, Carlsbad, CA, USA), and 10% fetal bovine serum. The cells were cotransfected with GFP-tagged and RFP-tagged plasmids using Lipofectamine 3000. After 24 h, the cells were fixed in 4% paraformaldehyde in PBS (pH 7.4) for 20 min at 25 °C, after which they were washed 3 times with PBS containing 0.1% Triton X-100 and blocked for 40 min with PBS containing 1% bovine serum albumin and 0.1% Triton X-100. Finally, the cells were washed with PBST three times, and the nuclei of cells were labeled with DAPI. Images were captured using a fluorescence microscope (Axio Vert.A1; Zeiss, Jena, Germany) and processed with ZEN 2 software (Zeiss).

### 4.7. BRET Assays

*DLG4* and *HSPA1A* cDNAs were cloned into vector pFN31K-Nluc-CMV-neo using Flexi^®^ vector (Promega) to generate N-terminal NanoLuc^®^ fusions. *ARC*^WT^ and *ARC*^w/o spectrin^ cDNAs were cloned into vector pFN21A-HaloTag-CMV-Flexi using Flexi^®^ vector to generate N-terminal HaloTag^®^ fusions. A NanoBRET™ assay was performed in white 96-well plates according to the manufacturer’s protocol (Promega). Briefly, both NanoLuc^®^ and HaloTag^®^ fusion vectors were cotransfected into HEK293 cells. After 24 h of incubation, cells were replated into white 96-well plates with or without NanoBRET™ HaloTag^®^ 618 ligand. Following 24 h incubation, NanoBRET NanoGlo substrate (Promega) was added at 0.1 μL/well, and readings were collected for 0.3 s at the NanoLuc^®^ emission (460 nm) and NanoBRET™ ligand emission (620 nm) levels to measure the donor and acceptor signal, respectively, using a VarioskanFlash (Thermo Scientific). BRET was calculated as the ratio of the emission at 620/460 nm. Each experiment was repeated at least six times.

## 5. Conclusions

Our top-down approach identified several ARC-regulated genes and proteins involved in the ECM, synaptic membrane, and heat shock protein family, and the association between ARC and these genes and proteins warrants further investigation. Nonetheless, the transcriptomic and proteomic profiles presented here for *ARC*-KO HEK293 cells contribute to our understanding of the mechanisms underlying the effects of ARC and the molecular pathways involved in the pathophysiology of schizophrenia.

## Figures and Tables

**Figure 1 ijms-23-04498-f001:**
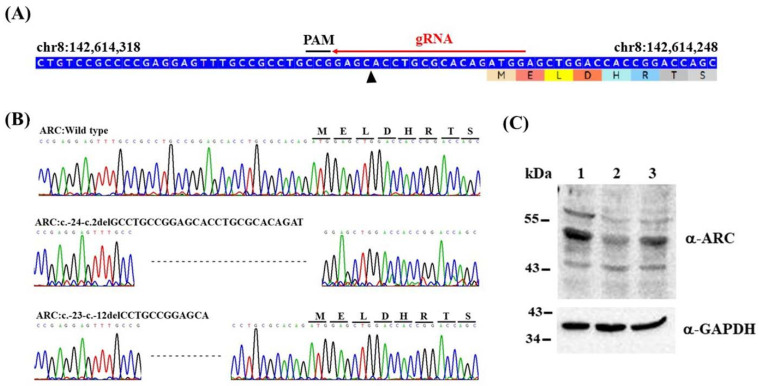
Generation and characterization of *ARC*-KO HEK293 cell lines: (**A**) Schema of the guide RNA target site of *ARC*. Arrowhead indicates the predicted double-strand break site: PAM, protospacer adjacent motif. (**B**) Sanger sequencing analysis of edited cell lines. (**C**) Representative immunoblots of ARC protein in wild-type (1) and edited (2: c.-24-c.2delGCCTGCCGGAGCACCTGCGCACAGAT; 3: c.-23-c.-12delCCTGCCGGAGCA) cells. GAPDH was used as the loading control.

**Figure 2 ijms-23-04498-f002:**
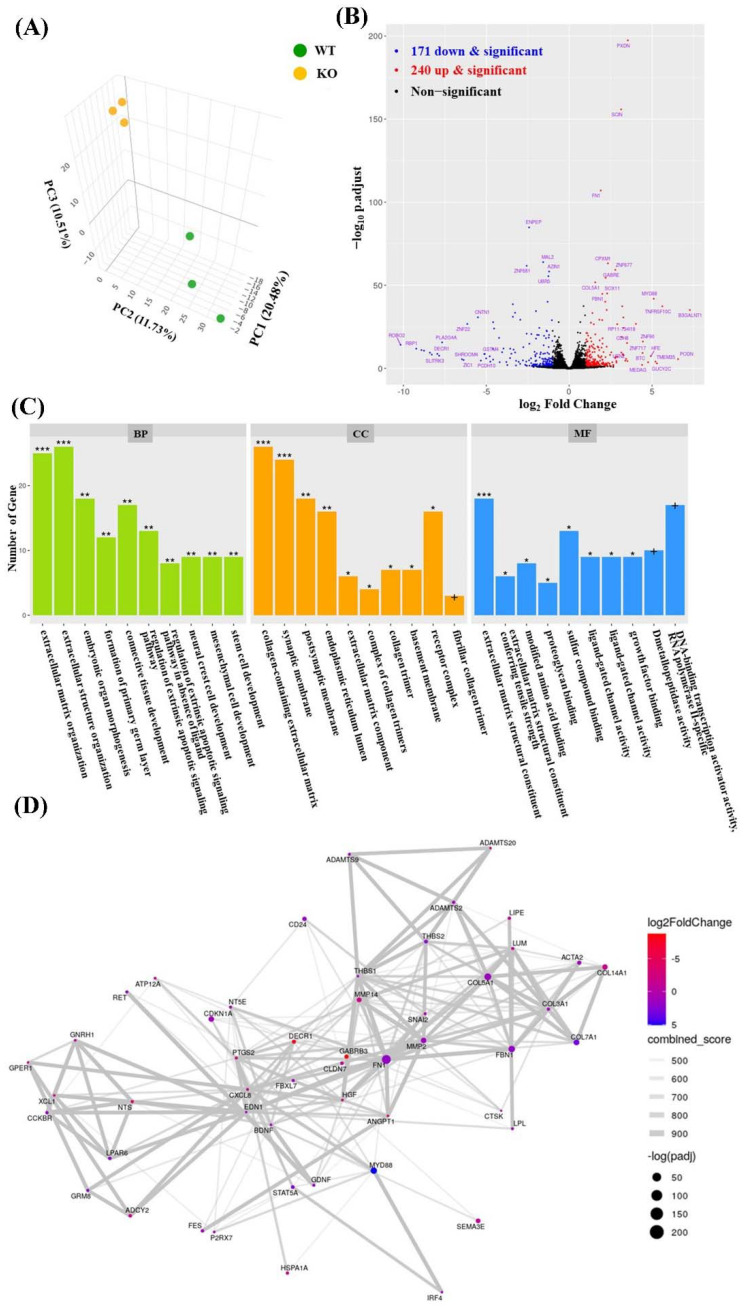
Analysis of RNA sequencing data: (**A**) three-dimensional principal component analysis plot; (**B**) volcano plot showing the DEGs between *ARC*-KO cells and *ARC*-WT cells; (**C**) GO terms associated with the DEGs in the biological process (BP), cell component (CC), and molecular function (MF) ontologies. Adjusted *p*-value, + < 0.1; * < 0.05; ** < 0.01; *** < 0.001; (**D**) protein–protein interaction network constructed using the top 50 DEGs.

**Figure 3 ijms-23-04498-f003:**
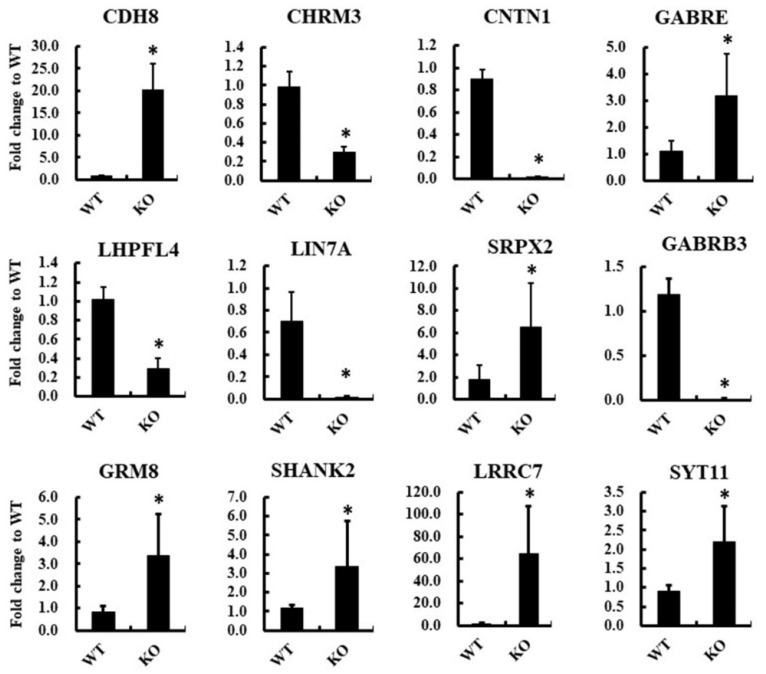
RT-qPCR assay results showing the expression of 12 synaptic membrane-associated genes (*CDH8*, *CHRM3*, *CNTN1*, *GABRE*, *LHFPL4*, *LIN7A*, *SRPX2*, *GABRB3*, *GRM8*, *SHANK2*, *LRRC7*, and *SYT11*) in *ARC*-KO and *ARC*-WT HEK293 cells. *GAPDH* was used as an endogenous gene for normalization. Data are expressed as fold changes relative to *ARC*-WT ± SD (* *p* < 0.05; *n* = 6).

**Figure 4 ijms-23-04498-f004:**
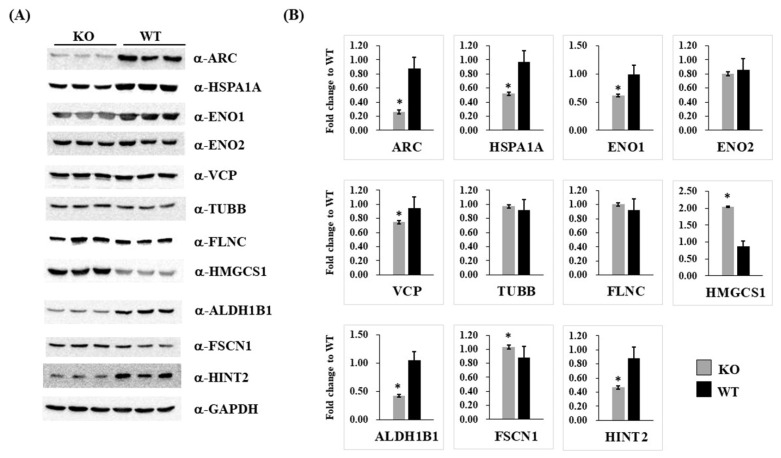
Immunoblotting analysis for validation of the differential expression of 11 proteins in *ARC*-KO cells. Immunoblotting (**A**) and quantification of protein expression (**B**) confirmed the differential expression of 11 proteins (ARC, HSPA1A, ENO1, ENO2, VCP, TUBB, FLNC, HMGCS1, ALDH1B1, FSCN1, and HINT2) in *ARC*-KO and *ARC*-WT HEK293 cells. GAPDH was used as the loading control. Data are expressed as fold changes relative to *ARC*-WT ± SD (* *p* < 0.05; *n* = 3).

**Figure 5 ijms-23-04498-f005:**
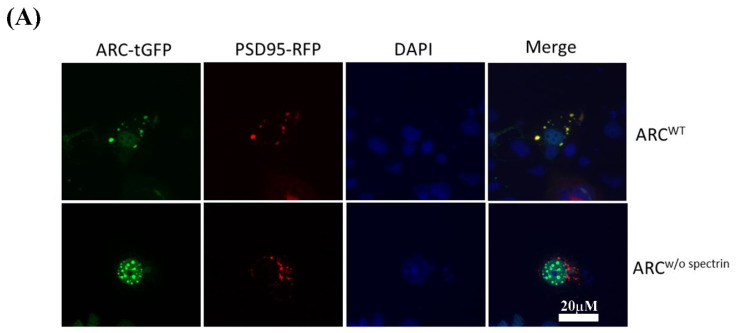
Protein–protein interaction analyses. ARC^WT^ (green) colocalized with (**A**) PSD95 (red) and (**B**) HSPA1A (red) in transfected COS-1 cells. (**C**) BRET assay analysis of ARC interactions in HEK293 cells. BRET ratios are shown for ARC^WT^/PSD95, ARC^w/o spectrin^/PSD95, ARC^WT^/HSPA1A, and ARC^w/o spectrin^/HSPA1A combinations, as well as the MDM2/p53 positive control. Each experiment was performed six times with (+) and without (−) the addition of the NanoBRET™ 618 fluorescent ligand. Data are means ± SD (*** *p* < 0.0005). DAPI (blue), 4′,6-diamidino-2-phenylindole.

**Table 1 ijms-23-04498-t001:** Top five GO terms enriched in *ARC*-KO-related DEGs.

Ontology	GO Accession	GO Term	Gene Count	Gene IDs	Adjusted *p*-Value
MF	GO:0005201	extracellular matrix structural constituent	18	*COL19A1*, *SRPX*, *SRPX2*, *COL7A1*, *FN1*, *COL21A1*, *PXDN*, *COL5A1*, *THBS1*, *LUM*, *ECM1*, *SBSPON*, *FBN1*, *COL3A1*, *PODN*, *THBS2*, *COL14A1*, *ZP3*	2.89 × 10^−6^
CC	GO:0062023	collagen-containing extracellular matrix	26	*COL19A1*, *ADAMTS2*, *MMP2*, *SRPX*, *SRPX2*, *SMOC2*, *COL7A1*, *FN1*, *COL21A1*, *PXDN*, *COL5A1*, *THBS1*, *LUM*, *ECM1*, *FLG*, *LAD1*, *ADAMTS9*, *SBSPON*, *FBN1*, *COL3A1*, *ADAMTS20*, *PODN*, *THBS2*, *COL14A1*, *ZP3*, *L1CAM*	7.29 × 10^−6^
BP	GO:0030198	extracellular matrix organization	25	*COL19A1*, *ADAMTS2*, *MMP2*, *ICAM2*, *NR2E1*, *SMOC2*, *COL7A1*, *FN1*, *PXDN*, *COL5A1*, *SPINK5*, *THBS1*, *LUM*, *CTSK*, *SCUBE3*, *GAS2*, *MMP14*, *CREB3L1*, *ELF3*, *ADAMTS9*, *FBN1*, *COL3A1*, *ADAMTS20*, *FOXC2*, *COL14A1*	7.29 × 10^−6^
BP	GO:0043062	extracellular structure organization	26	*COL19A1*, *ADAMTS2*, *MMP2*, *ICAM2*, *NR2E1*, *SMOC2*, *COL7A1*, *FN1*, *PXDN*, *COL5A1*, *SPINK5*, *THBS1*, *LUM*, *CTSK*, *SCUBE3*, *GAS2*, *MMP14*, *CREB3L1*, *ELF3*, *ADAMTS9*, *FBN1*, *COL3A1*, *ADAMTS20*, *LPL*, *FOXC2*, *COL14A1*	2.13 × 10^−5^
CC	GO:0097060	synaptic membrane	24	*CNTN1*, *LRRC7*, *LZTS1*, *SYT1*, *GABRE*, *SRPX2*, *DRP2*, *LIN7A*, *SLITRK3*, *SSPN*, *SYT11*, *CHRM3*, *FAIM2*, *CDH8*, *LHFPL4*, *GRIN2C*, *SHANK2*, *GPER1*, *BAALC*, *GABRB3*, *KCNJ4*, *NSG1*, *GRM8*, *GABRQ*	0.000229

MF, molecular function; CC, cellular component; BP, biological process.

**Table 2 ijms-23-04498-t002:** Summary of GO enrichment analysis in ARC-KO-related differentially expressed proteins.

Ontology	GO Accession	GO Term	Protein Count	Protein ID	Benjamini *p*
CC	GO:0005829	cytosol	54	OGFOD1, HMGCS1, GFUS, LSM1, RAD23A, SKP1, SEC31A, SET, UAP1, GALE, VPS29, ALDH1B1, ANXA1, ANXA5, ASNS, CAPN1, CAPZA2, CA2, CSE1L, CLTC, CPNE3, CAND1, CDK5, DCTPP1, ENO1, ENO2, EIF4G3, FSCN1, FABP5, FLNC, HSPA1A, HSPH1, PPA1, LONP1, MSN, MYH9, NUTF2, NUDT5, PGM3, PREP, PSMB3, PSMB6, PSMD5, PTMA, RPS18, RPS4X, RPLP1, RPLP2, SNRPD1, TLN1, TXNRD1, TUBB, YWHAB, VCP	1.06 × 10^−14^
CC	GO:0070062	extracellular exosome	34	GFUS, ANXA1, ANXA5, ANXA6, CAPN1, CAPZA2, CA2, CSE1L, CLTC, CPNE3, CAND1, ENO1, ENO2, FSCN1, FABP5, GLA, HSPA1A, HSPH1, PPA1, MSN, MYH9, NUTF2, NUDT5, PSMB3, PSMB6, RPS18, RPS4X, RPLP2, SERPINB1, TLN1, TXNRD1, TUBB, YWHAB, VCP	5.61 × 10^−12^
CC	GO:0005925	focal adhesion	16	ANXA1, ANXA5, ANXA6, CAPN1, CLTC, CPNE3, FLNC, HSPA1A, MSN, MYH9, RPS18, RPS4X, RPLP1, RPLP2, TLN1, YWHAB	1.17 × 10^−9^
CC	GO:0005737	cytoplasm	41	OGFOD1, HMGCS1, DDX39A, LSM1, RAD23A, SKP1, SEC31A, SET, ANXA1, ANXA5, ANXA6, CAPN1, CA2, CSE1L, CPSF6, CPNE3, CAND1, CDK5, ENO1, FSCN1, FABP5, FLNC, GLA, HSPA1A, HSPH1, HINT2, PPA1, ISOC1, MSN, MYH9, NOLC1, OAT, PREP, PSMB3, PSMB6, TLN1, TXNRD1, TUBB2B, TUBB, YWHAB, VCP	6.06 × 10^−6^
CC	GO:0005654	nucleoplasm	32	OGFOD1, DDX39A, RAD23A, SKP1, SET, UAP1, ALDH1B1, ANXA1, CSE1L, CPSF6, CPNE3, CAND1, CDK5, FABP5, HSPA1A, HSPH1, LONP, NUTF2, NOLC1, OAT, PUF60, PSMB3, PSMB6, PSMD5, PTMA, RPS18, RPS4X, SNRPD1, SF3A3, TXNRD1, TRIM28, VCP	7.81 × 10^−5^
MF	GO:0003723	RNA binding	20	DDX39A, LSM1, CLTC, CPSF6, CPNE3, ENO1, EIF4G3, FSCN1, HSPA1A, MYH9, NOLC1, PUF60, RPS18, RPS4X, SERPINH, SNRPD1, SLC25A5, SF3A3, TRIM28, VCP	2.92 × 10^−4^

## Data Availability

The raw data are available upon request of the corresponding author.

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
