# Peer review of "Transcriptomic and Proteomic Analysis of CRISPR/Cas9-Mediated ARC-Knockout HEK293 Cells"

_ijms, 2022, doi:10.3390/ijms23094498_

Round 1

Reviewer 1 Report

The paper describes the transcriptomic and proteomic analysis of CRISPR/Cas9-mediated ARC-knockout HEK293 cells. The steps taken were logical and easy to follow for the reader. My suggestion for the authors is since HEKs are used widely as a model cell line for the assay establishment it would be good to follow up these results in another stablished cell line/disease model.

Also the fonts for the figures need to adjusted as it is difficult to read in some like Fig.2 and 4. 

Reviewer 2 Report

Wang et al. submitted an research article on "Transcriptomic and proteomic analysis of CRISPR/Cas9-mediated ARC-knockout HEK293 cells". The authors produced an human ARC-KO HEK293 cell line using CRISPR/Cas9. The found that this cell line showed that genetic deletion of ARC disturbs several signaling pathways by using transcriptomic and proteomic analysis, including those related to the ECM, synaptic membrane, and heat shock protein family. The authors should explain why the HEK293 cell line was chosen to study the function of ARC, a kidney-derived cell type that may not reflect the involvement of ARC in regulating nerve cell function in psychiatric disorders.

1.In page 7, lines 153-155, the author described that the Immunocytochemistry revealed that ARC colocalized with PSD95 (Figure 5A) and HSPA1A (Figure 5B). But after careful comparison, it is found that they are not completely co-localized,the conclusions are overstated. The authors should explain why.

2.In Figure 5, DAP needs to be changed to DAPI, its full name should be provided as notes. How many repetitions were made?

3.Only bioluminescence resonance energy transfer (BRET) assays were used to conduct the protein–protein interaction analysis (Figure 5). The authors need to perform co-immunoprecipitation (Co-IP) analysis to further validate the interaction of ARC with PSD95, HSP1A1.

4.Transcriptome and proteome analysis data is not in-depth, especially how many differentially expressed proteins were found in this study? Are the two related?

Round 2

Reviewer 2 Report

The authors improved the manuscript after revision round 1. For me at this point the manuscript would be accepted.